# Assessment of an Assistive Control Approach Applied in an Active Knee Orthosis Plus Walker for Post-Stroke Gait Rehabilitation

**DOI:** 10.3390/s20092452

**Published:** 2020-04-26

**Authors:** Ana Cecilia Villa-Parra, Jessica Lima, Denis Delisle-Rodriguez, Laura Vargas-Valencia, Anselmo Frizera-Neto, Teodiano Bastos

**Affiliations:** 1Biomedical Engineering Research Group—GIIB, Universidad Politécnica Salesiana, Cuenca 010105, Ecuador; 2Postgraduate Program in Biotechnology, Federal University of Espirito Santo (UFES), Vitoria 29047-105, Brazil; 3Postgraduate Program in Electrical Engineering, Federal University of Espirito Santo (UFES), Vitoria 29075-910, Brazil

**Keywords:** active knee orthosis, biomechanical analysis, electromyography, gait rehabilitation, inertial sensors, rehabilitation robotics, stroke

## Abstract

The goal of this study is the assessment of an assistive control approach applied to an active knee orthosis plus a walker for gait rehabilitation. The study evaluates post-stroke patients and healthy subjects (control group) in terms of kinematics, kinetics, and muscle activity. Muscle and gait information of interest were acquired from their lower limbs and trunk, and a comparison was conducted between patients and control group. Signals from plantar pressure, gait phase, and knee angle and torque were acquired during gait, which allowed us to verify that the stance control strategy proposed here was efficient at improving the patients’ gaits (comparing their results to the control group), without the necessity of imposing a fixed knee trajectory. An innovative evaluation of trunk muscles related to the maintenance of dynamic postural equilibrium during gait assisted by our active knee orthosis plus walker was also conducted through inertial sensors. An increase in gait cycle (stance phase) was also observed when comparing the results of this study to our previous work. Regarding the kinematics, the maximum knee torque was lower for patients when compared to the control group, which implies that our orthosis did not demand from the patients a knee torque greater than that for healthy subjects. Through surface electromyography (sEMG) analysis, a significant reduction in trunk muscle activation and fatigability, before and during the use of our orthosis by patients, was also observed. This suggest that our orthosis, together with the assistive control approach proposed here, is promising and could be considered to complement post-stroke patient gait rehabilitation.

## 1. Introduction

The population that requires devices for rehabilitation and gait assistance has increased considerably due to the high numbers of cases of neurological impairments as stroke. Stroke is a leading cause of serious long-term disability, which is considered among the top 18 diseases contributing to years lived with disability [1]. Walking is more difficult for this group of people, as the resultant hemiparetic gait pattern after stroke is spatiotemporally asymmetric. Loss of balance, reduced propulsion at push-off, decreased hip and knee flexion during the swing phase, reduced stability during the stance phase of gait are typical, as well as a mixture of altered posture, and deviations and compensatory motion dictated by the residual function [2,3,4]. These conditions often lead to traumas, injury, disability, risk of falls, loss of independence, and a reduced quality of life [3,4].

Altered posture is normally related to muscle imbalances and altered joint position, which, ultimately, may result in movement dysfunction and pain [5]. A correct upright trunk posture can change muscle activation and improve the range of motion and symptoms [6]. Specific studies for the postural assessment of post-stroke patients have been conducted by [7], which includes items related to trunk control, such as sitting without support, standing with and without support, standing on the nonparetic leg, standing on the paretic leg, supine to affected side, supine to nonaffected side, supine to sit, sit to supine, sit to stand, stand to sit, and standing and picking up a pencil from the floor.

Computerized methods of postural assessment have been developed [8], ranging from digital images, force sensors on the body, and computer interfaces with information gathered from ‘wands’. Digital videography determines the patient’s posture through multiple images from the patient: force sensors on weight-bearing points at the soles of the patient’s feet allow infer posture; and wands analyze static standing posture of patients by placing the tip of the wand at various anatomical landmarks, whose data are processed into the computer, inferring patient’s posture. 

In this study, an innovative evaluation method related to the evaluation of patient’s dynamic postural equilibrium is conducted through inertial sensors. The method assumes that, in a neutral standing posture, the x-axes of the anatomical coordinate systems are aligned with the gravity vector. Then, through computing the orientation quaternions of each sensor to the initial technical-anatomical frames, the orientation of each body segment at any instant of time, and consequently the patient’s posture, can be obtained. This work also addresses the stance control, which is a strategy that can be used in rehabilitation tasks to reduce both energy expenditure and gait asymmetry (to both affected and unaffected legs), allowing less stressed paretic musculature in patients with muscular weakness [9,10]. This strategy also provides knee stability and protects the joint from collapsing during the stance phase of walking, releasing the knee to allow free motion during the swing phase [11].

### 1.1. Lower-Limb Powered Devices

Lower-limb powered devices have emerged for gait assistance and the rehabilitation of people with motor limitations [12], allowing them to walk and execute rehabilitation therapy [13]. These devices fall under two categories [12]: wearable joint actuators [14] and devices fixed to a platform (e.g., treadmill-based or paddle-based devices) [15]. Powered orthoses induce motion to one or more paralyzed lower-limb joints through external power (either electric, pneumatic, or hydraulic) [16]. Recently, exoskeletons have arisen as aid for over-ground bipedal ambulation, and a recent review was conducted by [17] about existing exoskeletons, including medical and non-medical assistive devices. Other studies have specially reviewed lower-limb exoskeletons in a clinical context, evaluating their good outcomes, such as effectiveness, possible benefits [18,19,20,21], but also their potential risks and adverse events, such as injuries on skin and soft tissue breakdown [22]. Regarding commercial exoskeletons, the U.S. Food and Drug Administration (FDA) has recognized them as Class II medical devices with special controls, clearing four of these exoskeletons for marketing in the U.S. [12]: ReWalk Personal (ReWalk Robotics, Yokneam Ilit, Israel), Indego (Parker Hannifin, Cleveland, OH, USA), Ekso GT (Ekso Bionics, Richmond, CA, USA), and Medical HAL (Hybrid Assistive Limb) (Cyberdyne, Ibaraki, Japan). In relation to powered lower-limb prosthetic devices, these are a relatively recent development, with only one commercially available powered ankle [23] and one powered knee [24]. Some of these robotic devices have been proposed in rehabilitation therapies for post-stroke patients and persons with paraplegia [21,25,26,27]. 

The two most common ways of controlling exoskeletons are direct force (or torque)-based control and position-based control [25,26]. However, a challenge in the field of robotic devices for rehabilitation, that still remains, is to design an effective control to maximize the benefits of these devices to the functional skills of patients [25,28]. The stance control strategy addressed here is considered as a new generation of controllers for orthotic intervention, which may potentially be significant to improve the patient’s gait kinematics [9] when also using an impedance control. 

### 1.2. Admittance Controller

An admittance controller is one variation of impedance controller, whose performance is determined by both force sensor and position. Compared to impedance control, admittance behavior is often more easily implemented in hardware [29]. A proper measure of the effectiveness of a system, which is meant to produce a rapid motion response to external forces, is the mechanical admittance *Y* [30], defined as:Y=v/F,
where v is the velocity of the controlled system at the point of interaction, and F is the contact force at that point. The basic idea of impedance control is to regulate the dynamic relation between the exoskeleton and the patient by relating the position error to the interaction force/torque through a mechanical impedance of adjustable parameters [26]. 

Due to the fact that humans change their joint impedances during walking by regulating their postures and muscle-contraction levels to maintain the stability, these kinds of controllers are of interest to develop such control strategies for gait rehabilitation devices [31]. The concept of impedance is intricately related to the mode and amount of muscle activation involved in the performance of a given task. Controlling the mechanical admittance of limb joints is an important feature of the neuromuscular system [32], as the output admittance and level of assistance are modified according to the patient disability level [31,33]. In fact, this is of vital interest for researchers involved with the design and control of prosthetic and orthotic devices [34]. 

The admittance control used here operates through the following first-order transfer function, relating input force with torque, inertia, and damping: (1)$$\overset{\cdot }{q_{a}}\left(s\right)=\tau G{\left(Ms+D\right)}^{-1}$$
where $\overset{\cdot }{q_{a}}$ is the output velocity, τ is the user-device interaction torque, *G* is the gain to modulate the admittance for each motion class, *M* is the inertia, and *D* is the damping. 

It is worth mentioning that, during gait, higher admittance values are needed at ground contact, whereas lower values are for swing phase, whose movements have great acceleration. For this purpose, a modulation method is employed to generate a suitable gain (*G*) to adjust *M* and *D* through information related to gait phases obtained from a footswitch insole [35]. The gait phases considered are as follows: initial contact (heel contact); mid-stance (flat foot contact); terminal stance (heel off); and swing (foot-off). More details about this control system can be found in our previous work [36], where also are shown validation results with volunteers. The results showed that this control system is able to execute movements according to the volunteers’ movement intention, providing suitable reference patterns for gait as well as proper torques for the motion classes considered here. 

Understanding how humans interact and adapt to robotic devices is critical to creating better designs for these devices [25], and even though a wide variety of controls are used, few works document their specific effects [25]. For instance, some studies related to the effectiveness of exoskeletons have investigated changes in gait parameters, using assessment metrics such as the distance walked, walking speed, isometric muscle strength, heart rate, respiration rate, blood pressure, and joint angle kinematics. Safety and usability evaluation with healthy subjects and post-stroke patients are also addressed [21,27,37,38,39]. On the other hand, studies conducted about muscles and inertial patterns during walking with these devices have focused on the evaluation of lower-limb muscle activity [36,37], with few studies evaluating other muscles related to maintenance of dynamic postural equilibrium during gait, such as trunk muscles [40]. It is worth commenting that the muscular activity gives a representation of which action users of these devices are attempting to execute. This activity may determine specific effects related to rehabilitation, such as compensatory movement, fatigue, and physical interaction between user and device. In fact, when patients wear a robotic device for mobility assistance, the movement pattern quality is completely dependent on their interactions with the device [25]. Thus, an evaluation of the activity of all the muscles that participate in the gait process in addition to kinematic information during walking are crucial to getting a successful optimization of the rehabilitation progress. 

The goal of this study is to assess the interaction between post-stroke patients and healthy people with an active knee orthosis plus a walker operating with stance control, during gait rehabilitation. The control runs an online method to modulate the knee impedance during gait, with the orthosis varying its knee support during walking, thus implementing a more natural gait pattern. In order to assess the user-orthosis interaction, kinematic and kinetic parameters are analyzed. In addition, an innovative evaluation of trunk muscles related to the maintenance of dynamic postural equilibrium during gait assisted by our active knee orthosis plus walker is also conducted. Gait-related muscle activity and gait trajectories are compared to both groups (post-stroke patients and healthy subjects). The following sections show details of our active knee orthosis, and its experimental implementation. The results and analysis of gait kinematics patterns and muscle activity of lower-limb and trunk muscles are also shown, which include aspects related to the orthosis safety and usability.

## 2. Materials 

### 2.1. Advanced Lower-Limb Orthosis for Rehabilitation (ALLOR)

ALLOR is the unilateral active knee orthosis addressed in this study, which was developed at UFES/Brazil [35,36] (Figure 1). The structure of ALLOR has a mechanical design that guarantees limits on the knee range of motion. Safety considerations (maximum angle and torque, and knee joint locked in the event of sensors failure) are incorporated in the control system, which are implemented in a PC104 and real time Simulink. ALLOR includes sensors to acquire signals from knee angle, knee torque, and plantar pressure to execute the control and to enable data analysis of the experimental trials. In our previous studies [35,36], ALLOR has been demonstrated to be useful for people with motor disabilities, providing them powered assistance in the sagittal plane to sit-to-stand, walk, knee extension during the swing phase of gait, and the flexo-extension of the hip and ankle. Details of the main components of ALLOR can be seen in Figure 1.

ALLOR is to be mounted on the user’s left leg and adapts to different anthropometric configurations, including heights from 1.5 to 1.85 m and weights from 50 to 95 kg. To ensure correct alignment during operation, a backpack and rigid suspenders are used. The backpack consists of shoulder straps and a belt that wraps around the user’s waist. The belt adjusts at the hip joint to support its structure. The backpack is used to house the surface electromyography (sEMG) electrodes, and there is also a free space to access them in the user’s lumbar area. The rigid straps are covered with a soft material, which can be adjusted through velcro straps to different diameters and lengths of the user’s leg. The total weight of ALLOR is 3.4 kg, including 0.8 kg of the backpack. The hip joint has a manual angle adjuster for flexion and extension from 0 to 80 degrees. Although this joint is not active, its regulation, according to the user’s requirements, allows establishing a safe range of motion.

The components of the active knee joint consist of a brushless flat motor (model 408057, Maxon Motors, Sachseln, Switzerland), a Harmonic Drive gearbox (model CSD-20-160-2A-GR, Harmonic Drive LLC, Peabody, MA, USA), and a servo motor (model AZBH12A8, Advanced Motion Controls, Camarillo, CA, USA). In addition, ALLOR is equipped with strain gauges (Wheatstone bridge configuration, model RS PRO Wire Lead Strain Gauge 3.5mm, RS Components Ltd., Corby, UK), which measure the torque produced by their interaction with the user. A precision potentiometer (model 157S103MX, Vishay Spectrol, Malvern, PA, USA) is used as an angular position sensor to measure the knee angles. In addition, a customized instrumented insole was built with four resistive force sensors (FlexiForce A401, Tekscan, Boston, MA, USA) to measure plantar pressure and recognize the gait phases. ALLOR also uses Hall-effect sensors inside its actuators to calculate their angular velocities.

The computer used to implement the ALLOR control software is a PC/104 (Advantech Co., Ltd., Milpitas, CA, USA), which is a standard for embedded computers. The electronic modules are composed of a motherboard, a power supply, an Ethernet communication and an analog-to-digital (A/D) acquisition card, model Diamond-MM-32DX-AT (32 16-bit inputs, 4 outputs 12 bit, and a maximum sampling rate of 250 kHz, Diamond Systems, Sunnyvale, CA, USA). All sensors, acquisition equipment and speed controller are connected via an A/D converter. The entire system requires a 24 V/12 A DC power supply and uses a CAN (Controller Area Network) bus operating at 1 Mbps. The control software was implemented in Simulink/Matlab (MathWorks, Natick, MA, USA) and uses a real-time library. Safety conditions were incorporated into ALLOR’s control system along with mechanical stops, which ensure that the actuator operates within the normal range of motion of the knee, allowing its safe use for the user.

The ALLOR’s control system is based on a hierarchical structure made up of a high-level Human Movement Intention Recognition (HMIR) system (the input of which comes from the movement intention captured through sEMG electrodes located on the user’s trunk). For translation of the user’s movement intention towards ALLOR movements, the controller includes, at the middle level, a Finite State Machine (FSM), which establishes the control strategy corresponding to the suitable gait movement. Finally, the admittance controller, speed controller, and Proportional Integral (PI) controller take care of performing the desired low-level motion, sending commands to the actuators, which move the ALLOR structure.

The FSM was developed in Stateflow at Simulink/Matlab, which is an environment for modeling and simulating sequential decision logic based on state machines. The FSM guarantees that the control system remains in a certain class, changing it only in case the HMIR output follows a logical sequence of movements.

The FSM is in charge of leading with the following motion classes: (1) Stand-Up (SU); (2) Sit-Down (SD); (3) Knee Flexion-Extension (F/E); (4) Walking (W); (5) Rest in Stand-Up Position (RSU); (6) Rest in Sit-Down Position (RSD). Then, for gait movements, the FSM is used to carry out the following transition sequence: RSU-SD-SU-W-RSU-SD-F/E-SU. Thus, once the HMIR system detects a movement intention, the FSM uploads the corresponding parameters of velocity $\overset{\cdot }{q}$ and admittance to activate the low-level controller. For instance, in SD and SU, $\overset{\cdot }{q}$ activates the PI (Proportional Integral) controller in order to let the joint execute the corresponding movement. Table 1 shows features of ALLOR regarding its structure and control levels.

### 2.2. Controller

The stance control strategy proposed here consists of the following features: (1) suitable free knee motion in the swing phase to allow free joint rotation in flexion and extension; (2) suitable lock of the knee joint to resist knee flexion while allowing free knee extension. This strategy was used in our previous study [36] for both body support and free leg movement, for different user’s anthropometry. Our controller has proved to be efficient, generating high and stable stiffness needed to avoid knee collapse during the stance phase, with rapid motion response to external forces. 

Here, a modulation through variable gain is used to increase or decrease the impedance components (damping and inertia), according to the gait sub-phases, in order to adapt the knee joint impedance during walking (see parameter values in [36]). The objective of this modulation is to allow the precise adjustment of knee impedance during gait cycle to obtain a smooth and quick switching between gait phases, making it possible for users to have both a fluid swing-phase movement and stance-phase stability. An algorithm is also used to recognize the other gait sub-phases: initial contact (defined by the heel contact); mid stance (defined by a flat foot contact); terminal stance (defined by the heel off); and swing (defined by the foot off based on ground reaction forces during gait) [36]. A new knee velocity pattern is also proposed here, which is based on knee impedance modulation, allowing body support in the stance phase of gait and free leg movement in the swing phase. The stance control strategy starts once a movement intention is detected. Thus, it assists the knee joint movement during gait, i.e., it provides knee support during RSU and allows free knee movement in RSD.

### 2.3. sEMG and Inertial Sensors

The signal acquisition equipment used in this study is BrainNet BNT-36 (EMSA, Rio de Janeiro, Brazil), which has ten bipolar channels sEMG acquisition and four multi-propose channels in addition to a band-pass filter from 10 to 100 Hz, a notch filter at 60 Hz, and a sampling rate at 600 Hz. Through this equipment, signals from sEMG, knee angles, and foot contacts were acquired (see Figure 1).

On the other hand, signals from six inertial sensors (IMUs—Inertial Measurement Units) are acquired and processed through Tech MCS Studio software (Technaid, Madrid, Spain). The output data provides spatial orientation (based on proprietary extended Kalman filtering) in the quaternion format at 50 Hz. The IMUs are placed on the subject’s pelvis and lower-limbs, in order to capture their bilateral movements during gait, in real time, through an hub (connected via USB to a laptop) to synchronize all sensors via CAN. The HUB allows its synchronization with the sEMG acquisition system (BrainNET). Data processing was conducted with MATLAB. 

## 3. Methodology

### 3.1. Subjects

An experimental protocol was implemented for both groups (control group and patients): eleven healthy subjects (aged between 22 to 38 years) without motor dysfunction that could interfere with walking patterns, and three post-stroke patients (1 female, 2 males, aged between 53 and 58 years, 9 to 21 months after the stroke episode). This study was approved by the Ethics Committee of the Federal University of Espirito Santo (UFES/Brazil), with number: 64801316.5.0000.5542. Written consent from each participant was obtained, and a physical therapist evaluated all patients with a Mini Mental State Examination test, in order to discard cognitive alterations that could influence the study.

### 3.2. Experimental Protocol

To conduct the experiments, the knee and hip of the patient were aligned with the rotation axes of ALLOR (assisted by a physical therapist). The acquisition of sEMG signals from trunk muscles was performed using BrainNET. Pairs of electrodes (Ag/AgCl adhesives, with 20 mm interelectrode distance) were used in a bipolar configuration, with fixation of the reference electrode in a local free of muscle fibers (in the right malleolus). The placement of the electrodes and preparation of the skin (trichotomy, abrasion and cleaning with 70% alcohol) was performed, following instructions provided by [41]. Measurements were made bilaterally in the following muscles: rectus femoris (RF), semitendinosus (S), and erector spinae (ES) at thoracic vertebra (T7), thoracic vertebra (T12), and lumbar vertebra (L4) levels. 

To capture the bilateral movements during gait, the data from the IMUs were acquired (see Figure 1a,b,d). The pelvis sensor was placed on the sacrum, at the S2 spinous process; the right and left thigh sensors were attached on the iliotibial tract, approximately 5 cm above the patella; and the right and left shank sensors were placed bilaterally, 5 cm above of the lateral malleolus of the fibula. The right foot sensor was fixed on the dorsal region of the foot on the 3rd and 4th metatarsal bones, 3 cm above the corresponding metatarsophalangeal joints. To avoid displacements, the IMUs placed on the pelvis, thighs, and shanks sensors were attached using 3D-printed supports fastened with elastic velcro strap. The group of patients used a four-wheel walker as a balance assistive device to ensure safety during experiments, and also to support both signal acquisition equipment (see Figure 1a).

To start the experiments, a synchronization signal (through a push-button) was sent to the equipment in charge of acquiring sEMG and inertial signals. After 2 s, each patient was asked to perform three walking trials in a distance of 10 m, in two different ways of walking: (i) without ALLOR; (ii) with ALLOR. Each user determined his/her suitable speed during the trials. 

### 3.3. Data Processing

In order to evaluate the sEMG and kinematics patterns during walking with and without ALLOR, the following analysis was performed:

#### 3.3.1. Bilateral Muscular Analysis and Fatigue 

We implemented an extended version of the method presented in [42] to estimate hip, knee, and ankle joint angles, bilaterally, using the pelvis inertial sensor as the reference to align the other sensors with the gravity vector and the walking direction. The first step consisted of calculating a correction quaternion to align the pelvis sensor’s x-axis with the gravity vector during the calibration process. The initial pelvis’ technical-anatomical coordinate system (i.e., the reference) is equal to the resulting quaternion after correction. Then, the initial technical-anatomical frames, which correspond to other body segments, are calculate based on that reference. The method assumes that, in a neutral standing posture, the x-axes of the anatomical coordinate systems are aligned with the gravity vector. Similar approaches were addressed in [43,44]. The second step consisted of computing the orientation of each sensor to the initial technical-anatomical frames, at the calibration instant, which is expected to be constant, as the sensors do not move with respect to body segments. Using these orientation quaternions, the orientation of each body segment at any instant of time was calculated as shown in (2):(2)q GcBi(t)=q GcSi(t) ⨂ (q BiSi)−1
where q represents the orientation quaternion, $B_{i}$ refers to the initial technical-anatomical frame of the *i* body segment, $S_{i}$ the sensor frame located on the *i* body segment, $G_{c}$ denotes the global coordinate system, and ⨂ is the Hamilton product. Finally, the joint angles of the patient are computed, relating the distal to the proximal body orientation quaternions. Details of these steps can be found in [42].

#### 3.3.2. Bilateral Muscular Analysis and Fatigue 

Raw sEMG data collected from each trial (walking without/with ALLOR) were pre-processed by a band-pass filter (Buttwerworth, 2nd order) from 20 to 100 Hz. Afterwards, these filtered signals were segmented for each gait cycle through the angular velocity provided by the inertial sensor placed on the left foot. Then, these segments were used to evaluate the bilateral activation between contralateral muscles on the trunk and lower limbs, as well as the muscular fatigue. 

For the bilateral analysis, the envelope of sEMG signals of each segment was computed, firstly through rectification, and after smoothing it by applying the root-mean square (RMS) over sliding windows of 100 ms. As a result, the mean bilateral ratio between contralateral muscles was obtained by computing the difference of the mean value obtained from the root mean square of the sEMG envelope corresponding to all gait cycles, as shown in (3):(3)$$BR_{k}\left(\%\right)=\frac{\sum_{k=1}^{N}A_{k}/N-\sum_{k=1}^{N}B_{k}/N}{max\left\{\sum_{k=1}^{N}A_{k}/N;\sum_{k=1}^{N}B_{k}/N\right\}}\times 100$$
where $BR_{k}$ is the mean bilateral ratio, $A_{k}$ and $B_{k}$ are the root square values computed over sEMG envelopes corresponding to both right and left muscles during the gait cycle k, and N is the total of the gait cycle. 

For the muscular fatigue analysis, the average of the median frequency and muscle activity ratio were calculated for each muscle. For this, two sets of filtered sEMG (X1 and X2) and two sets of sEMG envelopes (E1 and E2) were obtained from all gait cycles, taking in account the total cycles corresponding to the first (X1 and E1) and second half (X2 and E2) of the programmed pathway, respectively. Then, the fast Fourier transform (FFT) was computed on X1 and X2 in order to obtain, for each set, the average of median frequency. The muscular fatigue is detected by evaluating the decrease in the median frequency. Similarly, the muscle activity ratio was obtained from the ratio between the average of root mean squares (MeanRMS) for the last set and the first set, calculated on E2 and E1 as:(4)$$\mathrm{Muscle}\;\mathrm{activity}\;\left(\%\right)=\frac{{\mathrm{MeanRMS}}_{\mathrm{last}\;\mathrm{set}}\;}{{\mathrm{MeanRMS}}_{\mathrm{first}\;\mathrm{set}}}$$

### 3.4. Statistical Analysis

A Wilcoxon signed rank test and a Wilcoxon rank sum test were used to obtain a statistical comparison (*p*-value < 0.05), and the concordance correlation coefficient (CCC) was used to compare the gait pattern between the control group and patients, obtained from the inertial sensors, as done in [42]. CCC was used to know how much close, with respect to the control group, each patient reproduced joint angular variations during walking, being considered values from highly correlated to very highly correlated (*pc* > 0.80).

## 4. Results

Figure 2a–c show the results of the plantar pressure, gait phase, knee angle, and impedance modulation based on gain variation and knee torque when walking with ALLOR, for the post-stroke patients, which demonstrate the effects produced by the proposed stance control strategy. For all these cases, the gait phases were recognized using data from the plantar pressure. Regarding this, the results show a constant pattern of the plantar pressure for P1 and P2, which indicates that the actions of ALLOR did not cause variations on the foot pressure during the assisted gait. In addition, for all the patients, the flexor-extensor torque during the gait was approximately 5 Nm lower. 

In relation to the dominant laterality, no significant difference was detected in any of the patients, comparing their gait without ALLOR and with ALLOR (Figure 3). On the other hand, comparing the muscular action levels before and during the use of ALLOR, a significant difference was observed in the action of the ES muscle at T12 and L4, and in the S muscle of patient P1. In doing the same comparative analysis, patient P2 presented a significant difference in trunk muscle activation at T12. 

The results related to the kinematics patterns are shown in Figure 4, in which the gait analysis of the post-stroke patients was compared to the control group. Significant correlation of gait pattern at the right hip and knee without ALLOR in patient P1, at the left knee with ALLOR in patient P2, and at the left hip and right hip and knee in patient P3 were found, as shown in Table 2. On the other hand, a significant reduction in muscle fatigability was detected in patient P2, comparing his free gait to the gait using ALLOR. Also, a significant increase in fatigability was detected in patient P1 (as shown in Figure 3f). 

Regarding the patients’ satisfaction with the use of ALLOR, a survey was conducted in [36], using the Quebec User Evaluation of Satisfaction with Assistive Technology (QUEST2.0). Twelve questions evaluated the patients’ opinion regarding to dimensions, weight, adjustments, safety, durability, simplicity of use, comfort, and effectiveness of ALLOR. The score for each question ranges from 1 to 5 (1—“not satisfied at all”; 2—“not very satisfied”; 3—“more or less satisfied”; 4—“quite satisfied”; and 5—“very satisfied”). According to the patients’ feedback, their satisfaction with ALLOR (controlled by the proposed stance control strategy) was scored as (mean and standard deviation): dimensions: 3.91(0.00), weight: 3.91(0.82), adjustment: 4.22(0.94), safety: 4.64(0.47), durability: 3.91(0.82), ease of use: 5.00(0.00), comfort: 4.31(0.47), and effectiveness: 3.91(0.82), which can be considered very good scores for this first prototype.

## 5. Discussion 

When analyzing the data referring to the left hip flexion angles during gait with ALLOR (GWA) and gait without ALLOR (GWOA), patient P1 and P2 presented increased hip flexion and increased support time of the left lower limb during GWA. Moreover, a reduction in amplitude of this movement during the swing phase was verified in patient P3. The hip trajectory in the right side shows a difference in the stance phase for patient P2 when compared to GWOA. For the control group and patients P1 and P3, the hip does not show variations in the flexion-extension movement on the right side during both gait conditions, which indicates that ALLOR did not affect the hip movement of their lower-limb (which does not receive assistance from the orthosis). 

Regarding the knee joint, patients P1 and P3 increased their stance phase during GWA, and patient P1 had a marked reduction in his knee flexion during the swing phase. The results show a lower percentage of swing phase compared to the 40% of the gait cycle indicated for normal gait. Our results are corroborated by other studies ([37,38]) conducted with healthy subjects, which compared the gait with and without an exoskeleton, concluding that a gait assisted by robotic devices has an increased time in stance phase. However, patient P2 presented a knee angle most similar to the control group, and with a marked increase in flexion of this joint when analyzing the pattern of his left knee movement during GWOA. This implies a reduction in the stance phase, which is contrary to the other patients and to the aforementioned studies. In this sense, the fact that ALLOR is a unilateral orthosis can influence in the joint assisted, including the control group. It is clear that the use of an assistive device produces influence in gait phases, although more evaluations with patients are required. Despite this, the knee trajectories in the right side do not indicate compensations during the GWA. 

Regarding the ankle, some abnormalities were detected when comparing GWA to GWOA, as there was a change in the initial foot contact of patient P1, from heel to metatarsal region. Here, the angle of the control group also shows variations respect to both gait conditions, as patients P2 and P3 show angle variations in the swing phase that may reflect an adaptation of their joint due to the assistance applied on the knee by ALLOR. It is worth commenting that for cases where an assistance of the knee flexion or ankle dorsiflexion in swing phase during walking is required, a position control can be included at the beginning of this phase. For this, a preliminary evaluation of the patient is required to stablish suitable control strategies. 

Muscle fatigability is characterized as an inability of supporting muscle contraction for a period [45]. Regarding post-stroke patients, a study conducted by [46] evaluated electromyography in active-assisted exercises, detecting an increase in fatigability in some of their muscles required to carry out the tasks. Divergent results were observed about muscle fatigability in our study, through analysis of the results related to the sEMG, as a reduction in the muscular activation on trunk at the T7 vertebra level was observed, contralateral to the affected side, and a significant reduction in the muscle fatigability of the ES muscle at this level during GWA in patients P1 and P2. However, for patient P2, despite this reduction, there is a predominance of muscle activity on the non-affected side. In fact, as this muscle aids the trunk stabilization during the initial contact at gait phase, the increase in ES muscle activity of the non-affected side may have contributed for the lateral stability, which is a neuromuscular strategy that is normally adopted by hemiparetic post-stroke patients to reduce their body sway during walking without support, such as reported in [40]. 

At the T12 vertebra level, patients P1 and P2 presented an important synergy of bilateral trunk muscle activation using ALLOR, with a consequent reduction in muscle fatigability in this region. Patient P1 presented a reduction in activation in the ES muscle (left of the lumbar region), causing a greater bilateral balance in this region. Some studies [38,39] also found a reduction in lower-limb muscle activation in chronic post-stroke patients when using a robotic exoskeleton. This corroborates our results, as a reduction in activation in S muscle was found in patient P1 when using ALLOR. On the other hand, patient P2 presented a compensatory pattern of his hip flexion on the not-affected side during the initial support phase with ALLOR, which is consistent with the increase in the right S muscle fatigue on this side. This is consistent with the results of the hip angle, which show variations when comparing GWA and GWOA. 

Regarding the use of walkers with an exoskeleton for gait rehabilitation, in our study, the gait was assisted by a four-wheel walker, as it enhances the safety of the patients. However, in fact, this influences the muscle activity, especially in the trunk. In [40], a study analyzing the muscular activation of the trunk and lower limbs of healthy subjects using a lower-limb exoskeleton concluded that there are differences in walking with an exoskeleton and with a treadmill, as the former attenuates the movement efforts, due to its body weight support at the trunk level. 

In relation to the knee torque required for GWA, for all patients in our study, the torque decreased during the stance phase, which indicates that ALLOR offers suitable knee support. On the other hand, during the swing phase, the torque increases as expected; however, this does not overstress the joint. In [37], the knee torque during gait using a robotic device to assist users’ knee joints was approximately 2 Nm; however, the pelvis, hip, and foot were not attached to the device. This fact influences the knee torque measurements and demonstrates that a portable active orthosis requires suitable adjustment and low weight in order to reduce the knee torque during the swing phase. 

The muscular activity, kinematics, and kinetic measurements evaluated in our study indicate that the users (control group and patients) had a suitable interaction with ALLOR and with the stance control strategy based on impedance modulation. These results demonstrate the importance of evaluating the immediate effects of the active orthosis on the body alignment and muscle activity during gait, in order to plan a suitable rehabilitation program. 

In the survey conducted by [21], it was reported that there is little information regarding the impact of the use of a bilateral exoskeletons compared to unilateral devices for post-stroke gait rehabilitation. In our study, our preliminary findings indicate that the use of a unilateral robotic orthosis for the assistance of knee joint does not cause any movement compensation patterns able to affect the non-affected lower-limb during gait rehabilitation. However, the execution of a clinical protocol is fundamental to evaluate the effectiveness of ALLOR in post-stroke patients for the long term. This implies analyzing implications in gait rehabilitation, such as walking speed improvement, perceived exertion, subject satisfaction regarding pain levels, metabolic cost and limits of usage time, as reported in other studies [27,46]. In addition, to have a complete assessment of the kinetic data, an additional study to measure the knee and ankle torque during GWA and GWOA is necessary.

Finally, based on the acquired experience during this study, we realize the necessity of a physical therapist or assistant to mount our orthosis on the users and to align its structure correctly at their lower limb. This fact plays an important role in the user’s comfort and data acquisition quality during the execution of the gait sessions. 

It is worth mentioning that, different to other studies, such as, for instance, [40], where only healthy subjects were analyzed, we evaluated here two different groups: post-stroke patients and healthy subjects. A limitation of this study was the low number of patients, however, we believe that three patients can highlight the effectiveness of our orthosis. 

## 6. Conclusions

The results from our study show that the impedance modulation and stance control strategy proposed here allowed developing a satisfactory gait for post-stroke patients without imposing a fixed knee trajectory. The muscular activity, kinematics, and kinetic assessment of the participants of this research indicate that our assistive control approach may be promising and could be considered to complement post-stroke gait rehabilitation. Although a reduction in muscle fatigability was detected in lower-limb and trunk muscles, it is necessary to evaluate the additional effects of ALLOR during gait in order to analyze their implications in the rehabilitation. As future work, a clinical study for the long term, with a larger sample, is needed. 

## 7. Patents

The exoskeleton ALLOR patent, entitled “Unilateral Lower-Limb Robotic Exoskeleton for Knee and Human Gait Rehabilitation Therapy”, was submitted to the Institute of Technological Innovation—INIT, at UFES/Brazil, and is in the progress evaluation process.

## Figures and Tables

**Figure 1 sensors-20-02452-f001:**
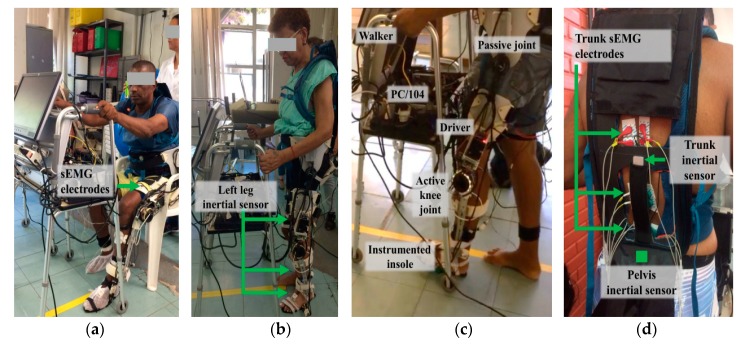
Patients accomplishing the experimental protocol with the exoskeleton plus walker, with surface electromyography (sEMG) and inertial sensors on their trunk. (**a**) sEMG electrodes location on the user’s leg; (**b**) Inertial sensors locations; **(c)** Location of the instrumented insole, active knee joint, driver, PC/104 and passive joint of the exoskeleton; (**d**) Location of sEMG eletrodes and inertial sensors on the user’s trunk and pelvis.

**Figure 2 sensors-20-02452-f002:**
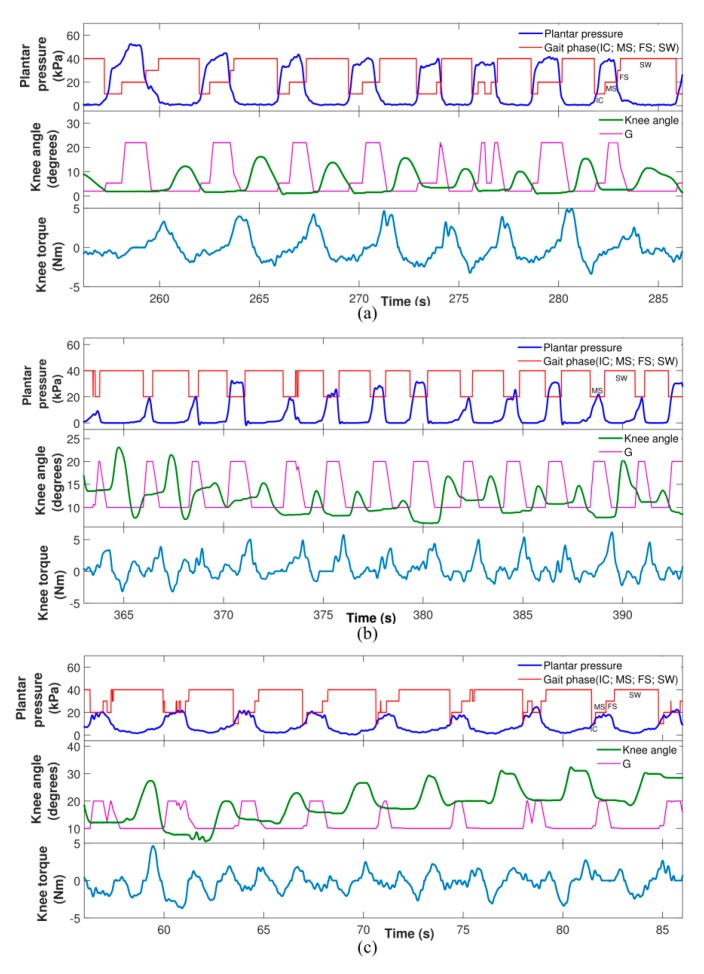
Example of acquired signals from the patients: plantar pressure, gait phase, gain variation (G), knee angle and knee torque. IC, initial contact; MS, mid stance; FS, final stance; SW, swing. (**a**) signals from P1; (**b**) signals from P2; (**c**) signals from P3.

**Figure 3 sensors-20-02452-f003:**
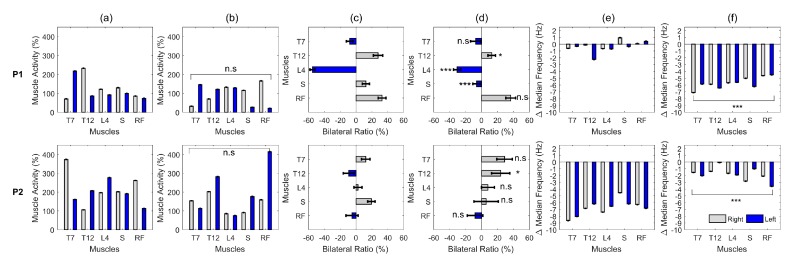
Effect of the use of the walker on the muscular activation of patients P1 and P2. (**a**) Muscle activation on different levels during gait without ALLOR; (**b**) with ALLOR; (**c**) Lateral synergy on muscle activation without ALLOR; (**d**) with ALLOR; (**e**) Muscle fatigability without ALLOR; (**f**) with ALLOR. n.s, non-significant difference; * *p*-value < 0.05; *** *p*-value < 0.001.

**Figure 4 sensors-20-02452-f004:**
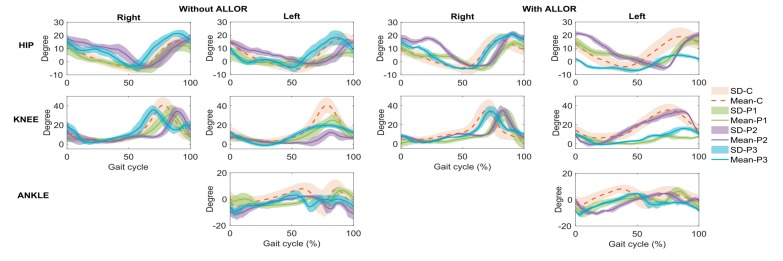
Kinematics patterns of the three post-stroke patients (P1, P2, and P3), obtained from inertial sensors, and compared to control group (C).

**Table 1 sensors-20-02452-t001:** Features of Advanced Lower-Limb Orthosis for Rehabilitation (ALLOR) regarding its structure and control levels.

**Weight**	3.4 kg
**User’s Heights**	1.5–1.85 m
**User’s Weights**	50–95 kg
**Control Levels**	High: Human Movement Intention Recognition (HMIR) through acquisition of sEMG from trunk muscles
	Middle: Finite State Machine (FSM) to switch the following classes of movement: Stand-Up (SU), Sit-Down (SD), Knee Flexion-Extension (F/E), Walking (W), Rest in Stand-Up Position (RSU) and Rest in Sit-Down Position (RSD)
	Low: Admittance Controller, Speed Controller, Proportional Integral (PI) Controller

**Table 2 sensors-20-02452-t002:** Concordance of joint angular variations into a gait cycle between each post-stroke patient and healthy control group.

Joint	P1		P2		P3	
	Right	Left	Right	Left	Right	Left
Without ALLOR						
**Hip**	0.91	0.49	0.51	0.52	0.78	0.83
**Knee**	0.82	0.77	0.43	0.24	0.77	0.72
**Ankle**		0.09		0.41		0.46
With ALLOR						
**Hip**	0.46	0.44	0.31	0.22	0.87	0.42
**Knee**	0.33	0.10	0.26	0.92	0.86	0.30
**Ankle**		0.33		0.09		0.34

P1: Patient 1; P2: Patient 2; P3: Patient 3.

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
