# Peer review of "Assessment of an Assistive Control Approach Applied in an Active Knee Orthosis Plus Walker for Post-Stroke Gait Rehabilitation"

_sensors, 2020, doi:10.3390/s20092452_

Round 1

Reviewer 1 Report

1、First of all, the content of the introduction part of the article is seriously insufficient. Some studies, such as clinical studies of the posture of patients with hemiplegia, and more robots with feedback should be detailed. In addition, it is recommended to give more introduction to "impedance" in the background section.

2、As shown in Figure 1, the design and structure of ALLOR were very complex. This posed burden and even the risk of tripping to all the participants. In addition, the use of a walker could also change the patient's gait and posture. This may interfere with the experimental results.

3、I think the "stance control strategy" mentioned in line 110 is the focus of this article. But, how did it work? What parameters was it controlled by? None of these key issues were made clear. These are conducive to the reader's understanding, which should be elaborated.

4、Also, as mentioned in line 110 of the article, I have doubts about whether the patients were really "suitable". It is necessary to add a questionnaire to assess the comfort of patients when using ALLOR. It would be helpful.

5、The algorithm, proposed to recognize the gait sub-phases, mentioned in line 112 may be problematic. Because the patient's gait is different from that of a healthy person, the method mentioned in this paper may lead to misjudgment.

6、As the author stated, only three patients participated in this experiment. The sample size is too small. Even worse, which side of the patient's body is sick and which side is healthy? This key data was missing from the article, which led to a significant reduction in the credibility of the results.

7、Although only three patients participated in the experiment, there were still inconsistencies in the results. This makes readers more confused.

8、Why did you use Wilcoxon signed rank test and Wilcoxon rank sum test to analyze the results? What is your statistical hypothesis? Is it non-normality? If so, reasons should be given.

9、Figures 2 and 3 in this paper are not clear enough. For example, readers may wonder what "G" in Figure 2 means. The reader may mess up the data of several patients in Figure 2.

10、There is an error in the mark number in line 180 and line231, where "3.3.1" is mistakenly written as " 3.1.1 " and "4" is mistakenly written as "5".

Author Response

Please, see the file attached.

Kind regards,

Reviewer 2 Report

  • Because of the long sentence and useless commas, grammatical errors occur, so replace it with a short and concise expression.
  • There is an error in the index, article 3.3.
    •  3.3 -> 3.3.1, 3.3.2, 3.3.3
    •  Duplicate index 3.3
  • Algorithm for impedance control and theoretical basis for control method to assist user behavior, and result of applied experiment are not shown at all, so it is unknown whether actual control for active orthosis is possible.
    • Where more accurate information is needed. 
      1. Specification of ALLOR
      2. Theoretical control algorithm and experimental results. (Response and control speed)

Author Response

(The authors gave the same response as above.)

Round 2

Reviewer 1 Report

According to the amendments proposed previously, the author modified the article very well. I think it meets the publishing requirements.

Author Response

Dear Reviewer,

We thank you for all your suggestions made during the review process!

Kind regards,

Reviewer 2 Report

There is still a lack of clarity on the physical properties, but many explanations have been added to make it understandable. However, it is considered to be easier to understand by arranging a table with respect to the applied control theory and the exact values of the parameters used (Line 172 ~ 217). It seems to have been well organized to some extent.

Author Response

Dear Reviewer,

Please, see the attachment!

Thank you!

Kind regards,
